# Application of Fluorescent Proteins for Functional Dissection of the *Drosophila* Visual System

**DOI:** 10.3390/ijms22168930

**Published:** 2021-08-19

**Authors:** Thomas K. Smylla, Krystina Wagner, Armin Huber

**Affiliations:** Department of Biochemistry, Institute of Biology, University of Hohenheim, 70599 Stuttgart, Germany; krystina.wagner@uni-hohenheim.de (K.W.); armin.huber@uni-hohenheim.de (A.H.)

**Keywords:** *Drosophila* eye, visual system, cornea neutralization, mosaic tissue, genetic screen, biosensor, GECI, calcium imaging, PI signaling, superresolution

## Abstract

The *Drosophila* eye has been used extensively to study numerous aspects of biological systems, for example, spatio-temporal regulation of differentiation, visual signal transduction, protein trafficking and neurodegeneration. Right from the advent of fluorescent proteins (FPs) near the end of the millennium, heterologously expressed fusion proteins comprising FPs have been applied in *Drosophila* vision research not only for subcellular localization of proteins but also for genetic screens and analysis of photoreceptor function. Here, we summarize applications for FPs used in the *Drosophila* eye as part of genetic screens, to study rhodopsin expression patterns, subcellular protein localization, membrane protein transport or as genetically encoded biosensors for Ca^2+^ and phospholipids in vivo. We also discuss recently developed FPs that are suitable for super-resolution or correlative light and electron microscopy (CLEM) approaches. Illustrating the possibilities provided by using FPs in *Drosophila* photoreceptors may aid research in other sensory or neuronal systems that have not yet been studied as well as the *Drosophila* eye.

## 1. Introduction

Fluorescent proteins (FPs) have revolutionized the localization of molecules within cells or living organisms. As compared to classical detection methods for localizing cell components, like antibodies for immunocytochemistry or DNA probes for in situ hybridization, recombinantly expressed FPs allow subcellular localization of molecules in real time. Depending on the system used, methods are available for non-invasive detection of FPs in living animals, plants or in cell culture cells. For instance, in a widely used approach, an FP like the green fluorescent protein (GFP) from jellyfish is fused to a protein of interest and recombinantly expressed in cell culture cells or transgenic organisms [1]. This allows subcellular localization of the protein of interest via its GFP fluorescence. The usage of FPs, however, is not limited to visualizing (fusion-)proteins. Numerous variants of FPs have been designed as biosensors to facilitate detection of, for example, specific membrane lipids (phosphoinositides) or Ca^2+^ ions, thereby expanding the usability of FPs beyond protein localization and addressing questions related to lipid signaling pathways or Ca^2+^ homeostasis.

The *Drosophila* visual system has helped to unravel the intricacies of light perception by phosphoinositide-mediated signaling and visual processing in neuronal circuits on a molecular level. In addition to investigations into visual processes like phototransduction and vision guided behaviour, the *Drosophila* eye has also been used as a model for membrane protein trafficking and neurodegenerative diseases. Neuronal cells are highly dynamic systems that need careful fine regulation of their homeostasis to function properly. One particularly complex topic pertains to proteostasis—the sum of all processes involved in protein biosynthesis, translocation, regulation and degradation—which is tightly linked to cellular health. Significant perturbation inevitably leads to the build-up of cellular stressors and ultimately results in cell death when a certain threshold is reached. Aside from developmental defects, two major components of cellular homeostasis have been proposed as etiological for retinal degeneration in the *Drosophila* visual system: Ca^2+^ control and Rh1 regulation [2]. Mechanisms contributing to cellular death include reduction, misfolding or mistrafficking of Rh1 as well as Ca^2+^-mediated excitatoxicity. Since rhodopsin is a highly abundant protein in photoreceptor cells (PRCs), even slight misregulation of its trafficking can result in rhodopsin accumulation with vast negative effects on the entire cell. In spite of anatomical differences, *Drosophila* and human eyes share a significant portion of conserved genetic pathways and degeneration causes [3]. Accordingly, understanding the link between proteostasis and neuronal degeneration in *Drosophila* may pose the foundation for future therapeutic treatments of human diseases.

Uncovering the details of neurodegeneration and its underlying mechanisms requires appropriate sensors and detection methods to match. Investigations of dynamic processes like vesicular transport, membrane protein trafficking or neuronal processing profit from live imaging setups. Like in many other fields, live imaging techniques of the *Drosophila* visual system for the most part rely on fluorescence for efficient visualization. Numerous genetically encoded reporters, subcellular markers and biosensors have been developed in the previous two decades on the basis of FPs. These have helped to screen for phenotypes, track subcellular translocation and detect changes in neuronal activity. In this review, we describe these various applications of FPs as examples for their usability in an intact and complex sensory system. Initially, though, we present the system of the *Drosophila* eye as well as some key techniques with which FPs are routinely detected in vivo.

### 1.1. Anatomy of the Drosophila Eye and Optic Lobe

The *Drosophila* compound eye is made up of several hundred unit eyes—called ommatidia—which are themselves each comprised of eight photoreceptor cells (PRCs, R1–8) as well as approximately 14 auxiliary cells, including cone cells, pigment cells and bristle complex cells (Figure 1A,B). Each PRC possesses a specialized plasma membrane—known as rhabdomere—which is made up of roughly 30,000 microvilli and acts as an optical waveguide housing the main proteins of the phototransduction cascade [4]. The rhabdomeres of the first six PRCs are arranged in a trapezoidal layout while the seventh and eighth rhabdomeres are located in the center and stacked on top of each other. A major component of these membranes are cell-specific light-absorbing rhodopsins (Rh), i.e., Rh1 in PRCs R1–6, Rh3 or Rh4 in R7 and Rh5 or Rh6 in R8. The *Drosophila* eye is routinely used for genetic screens and general studies regarding organ development, neurodegeneration and protein trafficking [5,6,7,8].

*Drosophila*’s visual system is divided into several interconnected layers of neuropils forming the optic lobe [9]. From distal to basal, the optic lobe consists of the retina (re), the lamina (la), the medulla (me) and the lobula complex, which can be subdivided into the lobula (lo) and the lobula plate (lop) (Figure 1B). There are two major types of neurons which either project within the optic lobe or from the optic lobe into the central brain. This system serves as a model for investigations into neuronal circuitry, sensory signal processing and visual perception [10,11,12].

### 1.2. Live Imaging of Drosophila’s Visual System

The regular and repeating structure of the *Drosophila* compound eye can be visualized microscopically by either autofluorescence of rhodopsins or by fluorescent labeling of any of the rhabdomeral proteins in combination with imaging techniques of *Drosophila*’s deep pseudopupil (DPP) and optical neutralization [13]. The DPP is a phenomenon of optical superposition owed to the exact and symmetrical orientation of the PRCs and ommatidia as well as the curvature of the eye and cornea (Figure 1C and Figure 2A). Since the layout of the PRCs within ommatidia is mirrored at the dorsoventral midline of the eye, the DPP is best seen when the microscope objective is aimed slightly above or below the eye’s equator. Optical neutralization, an alternative non-invasive method in this context, represents a form of microscopy in which the living animal is immersed into a medium (e.g., water) with a similar refractive index as the cornea and surrounding tissue to minimize light refraction and generate optical cross sections of ommatidia (Figure 2B,C). The ability to depict the retinal structure and detect intracellular fluorescence with comparably simple methods in a live imaging setup is one of the major advantages of the *Drosophila* visual system as a tissue for genetic screens and a model for neurobiological studies.

Investigations into neuronal activity within the optic lobe require more advanced methods. Aside from electrophysiological studies, which are for the most part confined to the outermost layers of the visual system, microscopic techniques have emerged here as well. However, imaging of living tissue is limited mostly by three factors: (1) poor resolution in the *z*-axis caused by excitation and emission outside of the focal plane, (2) shallow penetration of light due to scattering and (3) phototoxicity—the summation of negative side effects caused by prolonged irradiation of biological matter. While the first of these issues can be addressed optically, for example, by suppressing out of focus emission with confocal microscopy, the others remain. With the advent of two photon laser scanning microscopy (TPLSM), these restrictions had mostly been ameliorated to a point where in vivo imaging of brain tissue became possible [14]. In TPLSM, the energy required for reaching the excitation state of the fluorophore is provided by two photons which are absorbed almost simultaneously, namely within one femtosecond apart from each other. By carrying only half of the energy necessary for excitation, the photons can have longer wavelengths (800–1000 nm as compared to 390–600 nm in single photon excitation), therefore reducing scattering and phototoxic effects. As a side effect, optical sections are generated due to the fact that the only region where photon density is high enough to efficiently excite the fluorophore with two photons is at the focal point of the excitation laser. Accordingly, the resolution in the *z*-axis is significantly increased, comparable to a confocal microscope. Thus, in TPLSM of the *Drosophila* optic lobe, the exposed tissue of a living specimen is focused under a microscope using optical neutralization and excited by laser irradiation with wavelengths of the near infrared spectrum (Figure 2D). Recently, it has been shown that three photon excitation may even allow subcellular resolution imaging of the *Drosophila* brain through the intact cuticle [15].

## 2. Fluorescence Proteins in Forward Genetic Screens in the *Drosophila* Visual System

FPs serve a complementing dual role in targeted or untargeted genetic screens of the fly visual system: First, they visualize mosaic tissue in the investigation of lethal alleles by way of labeling genetically identical cell clones. Secondly, rhabdomeral proteins tagged with FPs present reporters for developmental defects, cellular integrity, membrane protein trafficking and more. Combined with DPP or optical neutralization microscopy, forward genetic screens in the fly eye are fast and allow efficient analysis of large libraries with relatively little effort in a short time span. Accordingly, this system has been utilized in one form or another for many screens as well as for targeted investigations in the past two decades, examples of which will be described in more detail below.

### 2.1. Using Fluorescent Proteins to Identify Cell Clones in Mosaic Tissue

The process of forward genetic screens has been exceedingly simplified with the inclusion of FPs. The fly eye serves as an excellent tissue for genetic screens due to its relative size and non-essential nature for adult viability. With regard to homozygous lethal mutations, which are difficult to investigate in adult organisms, mosaic tissue containing homozygous mutant cell clones is routinely generated. In this context, FPs function as a marker to visualize cell clones.

With the adoption of the yeast-based Flp/FRT system into *Drosophila*, induction of site-specific mitotic recombination became much simpler [16]. Via targeted exchange of two chromosome arms in heterozygously mutant cells during organ development, one of the emerging daughter cells may become homozygously mutant and grow into a mutant tissue patch. A set of fly stocks carrying centromere-adjoining FRT sites and an eye-specifically expressed flippase allow for readily generated adult mosaic eyes (Figure 3A) [17,18]. While the introduction of recessively cell-lethal genes onto the corresponding non-mutated chromosome arm results in the absence of homozygous wild type clones, the addition of an eye-specifically expressed apoptosis activator also eliminates heterozygous cells during eye development [18,19]. Thus, if required, the adult fly eye can almost entirely be made up of homozygously mutant cells which in most cases largely compensate for the lost tissue to generate a morphologically intact organ. On the other hand, the presence of immediately adjacent non-mutant clones as a reference may also be desirable in certain cases. Regarding the *Drosophila* eye, pigmentation appears to be the simplest way to distinguish mutant and non-mutant cell clones by utilizing a *white* (*w^+^*) marker in an otherwise *white* mutant (*w*^–^) background to generate red and white patches of retinal tissue. Unfortunately, due to their light-absorbing property eye pigments come with disadvantages for a number of downstream analyses, for example optical neutralization, DPP, immunohistochemistry, or assays involving the influence of light exposure stress on cellular integrity. This circumstance excludes numerous comparative investigations between mutant and non-mutant cells. Additionally, many transgenic constructs integrated into *Drosophila* stocks have traditionally been marked with some variant of the functional *white* gene, further limiting the utility of eye pigmentation in mosaic studies.

As an alternative to eye pigmentation, the Flp/FRT system has been expanded by adding ubiquitously or cell-specifically expressed FPs to mark clonal tissue [20,21]. These methods have later been improved upon into more advanced tools. Twin spot generator (TSG) introduced complementary GR and RG hybrid cassettes into the fly genome which combined the N-terminal portion of the green fluorescent protein (GFP) with the C-terminus of the red fluorescent protein (RFP) and vice versa [22]. The hybrid FP sequences are separated by an FRT sequence allowing mitotic recombination via flippase between the FP termini in heterozygous GR/RG cells to generate functional GFP and RFP genes in descended cells, respectively. Thus, TSG enables fluorescent tracing of both daughter cells from a specific division event. Meanwhile, Flybow uses two cassettes of two FP-encoding cDNAs (GFP/Citrine and Cherry/Cerulean), each arranged in opposing orientations and flanked by modified FRT sites [23]. Induced expression of an appropriately modified flippase during development causes inversion and/or excision events resulting in the expression of only one FP per cell, effectively labeling the tissue with four distinguishable fluorescent markers. In the *Drosophila* eye-specific tomato/GFP-Flp/FRT system, all R1-6 PRCs express a GFP::NINAC fusion protein localized in the rhabdomere (Figure 3B) [24]. Additionally, the red fluorescent fusion protein coding *tomato::NINAC* transgene is located on an FRT carrying chromosome arm, thus labeling the presence of a non-mutant chromatid. Accordingly, homozygous mutant tissue patches are identified by the expression of solely GFP which can be compared to GFP/tomato-expressing regions and even be traced over time due to the distinctive shape of the cell clones as visualized by fluorescent rhabdomere patterns. This property pairs perfectly with the non-invasive nature of DPP visualization and optical neutralization microscopy on living *Drosophila*’s eyes. Genetically distinguishable cell clone patches within the same individual also make it possible to differentiate between cell-autonomous and non-autonomous effects, while labeling techniques like TSG or Flybow facilitate effective lineage studies in neuronal tissue by utilizing numerous FPs simultaneously.

### 2.2. Using Fluorescent Proteins to Detect Structural and Protein Translocation Defects in Genetic Screens

Besides labeling of cell clones, FPs present reporters for photoreceptor cell integrity as well as protein localization in genetic screens. By way of labeling the repeating structure of ommatidia and the PRCs within them, FPs are highly useful to evaluate a possible impact of mutations on cell morphology, especially with respect to defects in the structure or spatial orientation of rhabdomeres. As reporters in the *Drosophila* eye, fusion proteins of GFP with different rhabdomerally located proteins such as rhodopsins, the ion channels TRP and TRPL, the signal regulator Arrestin 2 (Arr2) or the tail domain of the NINAC isoform p174 have been generated and expressed in PRCs (Figure 4) [25,26,27,28,29].

As sensory receptor cells, PRCs are highly specialized and precise regulation of their homeostasis is required in order to ensure proper functionality. This includes the composition of the plasma membrane and other membraneous compartments regarding lipids and proteins. Proteostasis is achieved by balancing protein synthesis, stability, degradation and targeted protein translocation. Trafficking of rhabdomeral plasma membrane proteins in *Drosophila* PRCs comprises the initial anterograde transport of newly synthesized proteins to the rhabdomere, endocytic internalization as a means of adaptation or signal deactivation, vesicular transport towards lysosomal protein degradation as well as various recycling processes [30]. Given that two of the membrane proteins of the phototransduction cascade in *Drosophila* (Rh1 and TRPL) undergo light-induced translocation between the rhabdomere and the intracellular space, fly PRCs offer a system in which neuronal membrane trafficking can be specifically triggered and thus systematically studied. While rhodopsin is extremely abundant in the rhabdomeral membrane, endocytosed in comparatively small numbers and largely channeled towards lysosomal degradation, virtually all TRPL molecules are endocytosed within several hours, remain stored in an intracellular compartment as long as illumination persists and are recycled back to the rhabdomere upon onset of darkness [31,32].

TRPL::GFP had been utilized in a genetic screen with the chemical mutagen ethyl methane sulfonate (EMS) to uncover components that are involved in membrane protein trafficking. This approach resulted in the isolation of 15 TRPL translocation defective (TTD) mutants and the detailed characterization of the P-Loop-containing, GTP- and phospholipid-binding protein TTD14 [28,33]. Following an RNA-seq screen, another study reported the importance of a CUB and LDLa domain protein (CULD) for endocytic turnover of both Rh1::GFP and TRPL::GFP within *Drosophila*’s PRCs [34]. Consistent with previous findings, defective protein translocation in *culd^1^* mutants resulted in retinal degeneration. Rh1::GFP was further used as a reporter in an RNAi screen for transcription factors regulating Rh1 expression and localization. In this screen, *ewg* was identified and described in its role in R8 photoreceptor subtype specification [35].

Evaluating reduced Rh1::GFP fluorescence in rhabdomeres also helped to identify mutants with defects in Rh1 homeostasis in several EMS-based screens. One such screen uncovered mutations in the DEAD box protein Dbp21E2 to affect Rh1 synthesis, maturation and trafficking which simultaneously resulted in Rh1-independent retinal degeneration [36]. Another study isolated mutants in V-ATPase V1 Subunit A1 which exhibit defects in post-Golgi trafficking of Rh1 and light sensitivity as well as synaptic dysfunction and degeneration of PRCs [37]. A third screen described mutations in Subunits 3, 5 and 6 of the ER membrane protein complex (EMC) as cause for Rh1 loss independent of the ERAD (ER associated degradation) pathway and retinal degeneration [38]. In combination with Flp/FRT mosaic eyes, low Rh1::GFP fluorescence in rhabdomeres was also an indicator of degeneration in a screen that identified phosphoethanolamine cytidylyltransferase (PECT) activity as essential for phosphoethanolamine homeostasis, phototransduction and neuronal cell integrity [39]. As an alternative read-out, the waveguide properties of the rhabdomeres themselves have been used in combination with optical cornea neutralization to infer defects in rhabdomeral organization caused by knock-down of genes potentially involved in vesicle trafficking in an RNAi screen [40].

Due to its physical interaction with metarhodopsin, Arr2::GFP has previously been used as an alternative to Rh1::GFP to investigate Rh1 transport defects by screening lethal transposon insertion lines as well as EMS-induced mutants via Flp/FRT mosaic eyes [41,42]. These screens resulted in the descriptions of a role for GPI-anchored proteins in post-Golgi cargo sorting processes towards the rhabdomere and the requirement of EMC Subunits 1, 3 and 8/9 for stabilization of immature Rh1 and other multi-pass transmembrane proteins, respectively. Later studies from the Satoh lab used the Arr2::GFP reporter to elucidate the importance of Rab6, its guanine exchange factor Rich and the SNARE protein Syx5 for crucial steps of intra-Golgi membrane protein transport as well as the involvement of the membrane curving F-BAR protein syndapin in segragating the rhabdomere from the stalk membrane [43,44,45].

### 2.3. Fluorescent Protein Fusions with Rhabdomeral Proteins in Investigations of Specific Mutants

The visualization of rhabdomeral proteins by way of fusion with FPs is also frequently used as a reporter in targeted investigations. For instance, the tomato/GFP-Flp/FRT system was used to demonstrate oppositional roles of tumor suppressor p53′s two isoforms regarding autophagy and apoptosis in one study, while another study showed that an eye-specific knockout of CSN4—a subunit of the COP9 Signalosome—resulted in a loss of photoreceptors which was not linked to apoptosis but rather proliferation impairment [46,47]. In another work regarding *Drosophila* as a model for Huntington’s disease, GFP::NINAC was expressed to visualize rhabdomeres of PRCs R1–6 and track age-dependent degeneration in HTT^97Q^ expressing PRCs by optical neutralization microscopy [48]. This study reported significantly accelerated neurodegeneration by protein phosphatase 1 (PP1) knock-down. Similarly, transgenically expressed Arr2::GFP served as a read-out for retinal degeneration in the phospholipase Cβ (PLCβ)-deficient mutant *norpA^P24^* [49]. This report demonstrated that while Rh1 phosphorylation was not required for interaction with Arr2::GFP, it appeared to be needed for internalization of Rh1/Arr2 complexes and thus to be partially responsible for retinal degeneration in *norpA^P24^* mutants. In a work utilizing the *Drosophila* eye as a model for Alzheimer’s disease, both Rh1::GFP as well as GFP::NINAC were utilized as rhabdomeral markers to record retinal degeneration [50]. Lambert et al. reported dysregulation of the endosomal trafficking machinery by the neuronally expressed isoform 1 of the genetic risk factor BIN1. Likewise, TRP::eGFP and optical neutralization is used routinely to investigate PRC integrity and determine or exclude degenerative processes from the experimental setups [28]. With respect to TRPL trafficking, FP fusions in chimeric TRP/TRPL ion channels were used to provide evidence for the necessity of both TRPL termini for light-induced protein translocation [51]. Furthermore, TRPL::GFP helped to uncover the involvement of RabX4 and Rab5 during light-induced TRPL internalization [52].

### 2.4. Other Genetic Screening Approaches in Drosophila Involving FPs

Apart from labeling of mosaic tissue and as reporters for cellular morphology or subcellular protein localization, FPs are utilized to determine gene expression. The integration of reporter constructs into genetic loci can be used to identify expression patterns of these loci and has come to be known in *Drosophila* as enhancer trap. This is based on the circumstance that transcriptional enhancers govern the expression of numerous genes in their sphere of influence which can encompass regions in the order of 10^5^ basepairs [53]. This then includes transgenes that are integrated via transposons into these regions of regulation within an organism’s genome. Among the first enhancer trap screens in *Drosophila* was the identification of photoreceptor-specifically expressed genes via FP reporters [54]. The addition of splice acceptor and splice donor sites flanking the FP sequence gave rise to protein traps which potentially generated whole libraries of *Drosophila* stocks expressing FP-tagged fusion proteins under endogenous regulation as long as the integration event took place within a gene intron. The Carnegie and *pigP* collections, for instance, are libraries of *Drosophila* stocks that carry insertions of either transposable P element or *piggyBac* cassettes containing the *white* or the *yellow* marker gene as well as the coding sequence of GFP or YFP, respectively, in hundreds of individual gene loci [55,56]. The MiMIC library similarly utilizes a *Minos* transposon cassette for genomic integration into various genes in combination with the GFP transgene [57]. Another project generated hundreds of fly stocks expressing C-terminally FP-tagged proteins by integration of cDNA fosmid libraries via the site-directed phiC31 system to study expression patterns and subcellular protein localization [58]. Together with further techniques, these libraries can be used in a variety of screening setups. Thus, iGFPi (in vivo GFP interference), for instance, specifically knocks down expression of GFP-trapped proteins by targeting the GFP-encoding portion of the respective mRNA through RNAi [59]. Further, deGradFP (degrade GFP), on the other hand, utilizes the highly conserved eukaryotic ubiquitin pathway for targeted proteasomal degradation of FPs and FP fusions via genetically encoded anti-GFP nanobodies [60]. Meanwhile, GrabFP and equivalent techniques use similar alpaca-derived nanobodies raised against GFP to force protein interactions and relocalize GFP-tagged target proteins [61,62]. Conveniently, while being the target of these approaches, GFP fluorescence simultaneously serves as reporter for successful knock-down or mislocalization. More recently, the exploitation of the bacterial CRISPR/Cas9 system as well as the ribosomal skipping peptide T2A has expanded the toolkit even further with, for example, Trojan exons or CRIMIC in which the integrated cassettes can easily be exchanged for constructs with various other properties [63,64]. A Cas9::T2A::GFP construct has also been implemented as a positive fluorescence marker in a novel CRISPR approach to identify Cas9 expressing cells in which a knock-out could be expected [65]. Raghu Padinjat’s group then proceeded to screen for regulatory genes of the phosphatidylinositol signaling pathway necessary for eye development.

## 3. Fluorescence Proteins in Functional Studies of the *Drosophila* Visual System

*Drosophila* phototransduction has been studied in detail as a neuronal model for G-protein coupled signaling pathways. Here, signal transduction starts with photons entering the compound eye through the cornea of an ommatidium. Once at the rhabdomere of a PRC, most of the light rays are assumed to be guided along the entire length of the rhabdomere by total reflection. This leads to a very high probability of photon absorption by abundant rhodopsin molecules embedded in the microvillar membranes. Fly rhodopsins are comprised of the opsin protein and the chromophore 11-*cis*-3-hydroxy-retinal which photoisomerizes into an all-*trans* conformation upon absorbing a photon. Photon absorption results in the formation of activated rhodopsin referred to as metarhodopsin which sets the phototransduction cascade in motion. Metarhodopsin’s interaction with the heterotrimeric G protein (G_q_) causes the exchange of GDP for GTP, which in turn leads to the release of the GTP-bound α-subunit of G_q_. G_q_α-GTP activates PLCβ which then cleaves the membrane lipid phosphatidylinositol (4,5)-bisphosphate [PI(4,5)P_2_] into the products diacylgycerol (DAG) and inositol (1,4,5)-trisphosphate (InsP_3_). Cleavage of PI(4,5)P_2_ also generates protons, measurable changes in the microvillar structure and is responsible for gating the ion channels TRP and TRPL in the rhabdomeral membrane, resulting in Na^+^ and Ca^2+^ influx and depolarization of the PRC. Changes in Ca^2+^ concentration then feed back onto numerous regulatory mechanisms, for example inactivation of rhodopsin, positive and negative regulation of TRP channels as well as short- and long-term light-adaptation [66]. The deactivation of the cascade as well as conservation and replenishing of the PI(4,5)P_2_ pool via negative feedback, the phosphoinositide cycle and vesicular membrane traffic are subject of ongoing research. Methods for tracking PIs and Ca^2+^ are accordingly of great importance in this field and will be discussed in this chapter.

### 3.1. Fluorescence-Based Calcium Sensors

In neurons, regulation of Na^+^, K^+^ and Ca^2+^ flux across the plasma membrane results in cellular de- and hyperpolarization as a response to external stimuli. Signal transmission at the synapse is then triggered by the Ca^2+^-mediated release of neurotransmitters. In addition, Ca^2+^ feeds back on neuronal signaling cascades in various ways. Accordingly, investigations of neuronal activity and function demand determination of cellular Ca^2+^ levels and its regulated changes in a time resolved manner. Apart from electrophysiological methods to study neuronal function, the establishment of appropriate Ca^2+^ biosensors for imaging has been vital to neurobiology. In addition to chemical indicators which are based on Ca^2+^-chelating agents that fluoresce, protein-based indicators have been developed over the years. In initial approaches, the bioluminescent protein Aequorin from the eponymous jellyfish *Aequorea victoria* was used as a sensor for Ca^2+^ ions, since its ability to luminesce is linked to Ca^2+^-binding. However, Aequorin’s luminescence also relies on a prosthetic group—the luciferin coelenterazine—which is absent in most biological systems. Therefore, a major advance has been the introduction of Ca^2+^-sensing molecules based on FPs referred to as GECIs (genetically encoded calcium indicators) which have several advantages over previous chemical indicators as well as Aequorin (Figure 5A). Most remarkably, these sensors allow the tracking of changes in intracellular Ca^2+^ concentrations in live imaging setups. The first iteration, Cameleon, implemented a pair of complementary FPs (CFP/YFP or BFP/GFP) linked by Ca^2+^-binding calmodulin (CaM) and M13, the C-terminal fragment from myosin light chain kinase, to generate the prototypical and ratiometric Ca^2+^-sensing Förster resonance energy transfer (FRET) probe [67]. Alternatives like camgaroo and G-CaMP used only a single circularly permuted (cp) FP in combination with CaM-M13 [68,69]. To account for fast dynamics and low levels of Ca^2+^ changes of neuronal cells, a mutagenesis approach coupled with in vivo evaluation has ultimately yielded the variant GCaMP6f with improved sensitivity for cytosolic measurements [70]. More recently, ER-GCaMP6–150 was developed with reduced Ca^2+^-binding affinity, an N-terminal signal peptide of ER-located calreticulin and a C-terminal KDEL retention motif to examine ER Ca^2+^ dynamics [71]. Ratiometric FRET probes were optimized as well by reducing the number of Ca^2+^-binding sites from four to two or even one in so called Twitch sensors in which the CaM-M13 domain was exchanged for a truncated troponin C (TnC) from the toadfish *Opsanus tau* [72]. To address the issue that red-shifted FPs require two photon laser excitation with wavelengths outside the near-infrared region (>1000 nm), the long Stokes shift Ca^2+^ sensor REX-GECO1 has been engineered [73]. REX-GECO1 can be excited by one 480 nm photon or two 910 nm photons and has been demonstrated to act as both an excitation as well as an emission ratiometric single FP-based Ca^2+^ probe [74].

### 3.2. Calcium Imaging in Drosophila Photoreceptors

In pioneering works, two groups used chemical Ca^2+^ sensors and electrophysiological recordings in parallel to study light-triggered Ca^2+^ changes in *Drosophila* PRCs [75,76]. These studies revealed that Ca^2+^ changes are spatially localized to the rhabdomeres where the light-dependent Ca^2+^ influx generates micromolar range transients essential for negative regulation of TRP channels. With respect to FP-based Ca^2+^ sensors for fast kinetics, GCaMP6f was the first to be employed in *Drosophila* PRCs. Utilized under regulation of either the synthetic eye-specific enhancer *glass* multiple reporter (GMR) or the Rh1 promoter, GCaMP6f has been used to characterize light-triggered Ca^2+^ changes with high precision, demonstrating its practical performance in the fly eye [77,78]. Since all GFP-based in vivo live imaging methods have to deal with light scattering and photon absorption by tissue as well as phototoxic effects at least to some degree, longer wavelengths with less energy are usually preferred. With regard to GECIs, red-shifted variants like RCaMP and R-GECO have been developed to address this issue and at the same time enable two-color fluorescence setups [79,80]. An enhanced variant of R-GECO with sensitivity comparable to GCaMP6f has been successfully used in photoreceptors of transgenic flies in combination with ER-GCaMP-150 to simultaneously trace complementary changes of cytosolic as well as ER Ca^2+^ levels and demonstrate that the Na^+^/Ca^2+^ exchanger CalX functions as a Ca^2+^ transporter in the ER membrane [81,82]. The *Drosophila* visual system has also been recently established as a model to investigate contractile forces in tissue morphogenesis [83]. Ready and Chang utilized the eye-specifically expressed GCaMP6m variant for intermediate kinetics to detect phototransduction-independent waves of Ca^2+^ propagating through interommatidial cells which promote stress fiber contractions that generate forces needed to shape the eye’s curvature.

### 3.3. Monitoring Calcium and Neuronal Activity in the Drosophila Optic Lobe

Sensing neuronal activity by light-microscopical means in *Drosophila* is not limited to Ca^2+^ changes within retinal neurons but can also be performed by visualizing signaling in the entire optic lobe—especially in combination with multiphoton imaging techniques which are able to use longer wavelengths for excitation and thus penetrate deeper into the tissue with little scattering. Accordingly, the FRET probe Twitch-2C was successfully utilized in a study uncovering color processing by inhibitory synaptic interactions involving histamine receptors HisCl1 and Ort in photoreceptor terminals of R7/R8 pairs [84]. Another recently published report compared measurements in the medulla obtained from GCaMP6f with those acquired via the glutamate-sensing reporter iGluSnFR which had been developed earlier by coupling a glutamate-binding GltI domain from an *E. coli* ABC transporter with a cpGFP (Figure 5B) [85,86]. The study presents iGluSnFR to have an even faster response than the Ca^2+^-sensing GCaMP6f in *Drosophila* vision, while admitting certain limits of this sensor regarding ligand diffusion due to iGluSnFR’s localization in the plasma membrane facing the extracellular space within the synaptic cleft. A further technique to image neuronal activity exists in the form of so called GEVIs (genetically encoded voltage indicators) which use a voltage-sensing domain—mostly from voltage sensitive phophatases of the tunicate *Ciona intestinalis* and chicken (*Gallus gallus*)—or opsins from various microbial species linked to a fluorescent reporter (Figure 5B) [87]. As with GECIs, some sensors facilitate FRET while others are based on modifying the properties of singular (cp)FPs [88]. Applied to medulla intrinsic and transmedullary neurons of the *Drosophila* visual system, concurrent measurements with GCaMP6f and the GEVI ASAP2f demonstrated that unlike voltage responses which consistently decay from the site of synaptic input, Ca^2+^ dynamics are compartmentalized and differ unpredictably between distinct regions [89]. This finding led Thomas Clandinin’s group to the interpretation that these neurons transmit specific signals to their downstream connections in different synaptic layers.

### 3.4. Phosphoinositide Tracking by Fluorescently Tagged Lipid Probes

The history of investigations into PI signaling is at least in part rooted in neurophysiological experimentation and as thus its importance for neuronal cell function as well as a role in disorders of the nervous system has long been established [90]. Emphasis has also been put on the role of phosphoinositides in the vertebrate and invertebrate visual systems [91,92]. Phosphatidylinositol (PI)—the basic molecule of this lipid family—can be phosphorylated on Positions 3, 4 and 5 of its inositol ring and all possible combinations thereof, leading to the generation of seven phosphorylated variations. As phospholipids, PI and its phosphorylated derivatives—known overall as phosphoinositides—are an integral part of eukaryotic membranes. Phosphoinositides are to various degrees participating in important cellular processes like signal transduction, cytoskeletal organization and vesicular transport. Presence of various phosphoinositides has also been described as characteristic for certain subcellular membranes. To unravel the plethora of mechanisms PI signaling is involved in, effort has been put forward to accomplish intracellular tracking of specific phosphoinositide levels and dynamics. Thus, in order to complement biochemical detection methods, which are not part of this review, fluorescent probes specific to either a single phosphoinositide or a subset of these lipids have been developed (Figure 5C). One of these probes comprised a fusion of GFP with the PI(4,5)P_2_-binding pleckstrin homology (PH) domain of the PI(4,5)P_2_-cleaving enzyme phospholipase Cδ (PLCδ) at either its N- or C-terminus to generate PH::GFP and GFP::PH, respectively [93,94]. However, these PH domains are not exclusively binding PI(4,5)P_2_ and also recognize the cleavage product and second messenger inositol (1,4,5)-trisphosphate (InsP_3_). Another phosphoinositide sensor for the early endosomally enriched PI(3)P was established in the form of GFP::2xFYVE, containing two zinc finger domains from tyrosine kinase substrate Hrs [95,96]. Subsequent endeavors yielded less ambiguous sensors for PI(4,5)P_2_ based on the PH domain of the murine transcription factor Tubby which was fluorescently tagged with either GFP or YFP [97,98]. A genome-wide analysis of PH domains in yeast lead to the generation of an entire set of GFP-fusions—both specific and promiscuous in their affinity towards different phosphoinositides, whereas FP-tagging of the PH domains from human OSBP and FAPP1 as well as the P4M domain of the secreted effector protein SidM from *Legionella pneumophila* resulted in PI(4)P-specific probes [99,100,101]. More elaborate ratiometric sensors were introduced to monitor phosphoinositides based on FRET and dimerization-dependent FPs [102,103].

### 3.5. Monitoring Phosphoinositides in Drosophila Photoreceptors via Fluorescently Tagged Lipid Probes

Given that *Drosophila* phototransduction utilizes a phosphoinositide-mediated signaling pathway, usage of phosphoinositide-specific fluorescent probes to study this system has yielded fruitful approaches. While the majority of studies that produced and characterized phosphoinositide sensors were based on cultured cells, a subset thereof (PH::GFP, Tubby^R332H^::YFP, P4M::GFP, OSH1::GFP and OSH2::GFP) has been integrated into in vivo investigations of *Drosophila* PRCs to test sensor performance regarding the monitoring of specifically PI(4,5)P_2_—arguably the most important of the phosphoinositides for PLC-mediated signal transduction—and PI(4)P [104]. In another study, Roger Hardie’s group later on tracked PI(4)P and PI(4,5)P_2_ with P4M::GFP and Tubby^R332H^::YFP, respectively, in recordings of *Drosophila*’s DPP in order to screen candidate genes for their involvement in phosphoinositide turnover and ultimately described PI4KIIIα as the kinase responsible for PI(4)P synthesis in *Drosophila* photoreceptors [105]. PI4KIIIα’s importance for PI(4)P as well as PI(4,5)P_2_ homeostasis and phototransduction was simultaneously corroborated by Raghu Padinjat’s group via similar experiments with P4M::GFP and PH::GFP [106]. Furthermore, detection of PI(4,5)P_2_ via PH::GFP in optical neutralization microscopy supported the claim that the kinase PIP5K was required for resynthesis of the PI(4,5)P_2_ pool following PLC-mediated depletion during phototransduction [107]. Aside from studies into the metabolic phosphoinositide cycle, fly PRCs are also utilized to investigate membrane contact sites between the endoplasmic reticulum (ER) and the plasma membrane (PM), in particular with regard to phosphatidylinositol transfer proteins (PITPs) [108]. By using PH::GFP to detect PI(4,5)P_2_ in the DPP, it was demonstrated that the PITP RDGB (retinal degeneration B) required an FFAT motif and two C-terminal domains (DDHD and LNS2) to interact with the ER integral protein dVAP-A as well as activity of an enhanced synaptotagmin (Esyt) to correctly localize to the ER/PM contact sites [109,110,111]. Fluorescently tagged phosphoinositide-binding domains additionally serve another purpose in cell biological questions. Due to PI(3)P’s significant enrichment in early endosomal structures, GFP::2xFYVE is routinely used as a subcellular marker, especially for studies regarding vesicular trafficking including investigations in the *Drosophila* eye [50,112].

## 4. FP-Based Tags and Sensors of Tomorrow’s Investigations

Arguably one of the most significant advances regarding microscopic imaging techniques in recent years was overcoming Ernst Abbe’s limit for optical resolution by several independent methods generally referred to as super-resolution microscopy or nanoscopy [113,114]. All of these microscopical approaches rely on certain properties of fluorophores. Single molecule localization microscopy (SMLM) techniques circumvent the diffraction limit of light by detecting only small subsets of all fluorescent molecules within the sample at a time. Therefore, the signals are effectively being spread out into fluorescent point sources before their visualization. An algorithm then utilizes the microscope’s point spread function (PSF) and mathematically infers from the Gaussian distribution a more precise localization for the origin of each individual fluorescence signal. One of these methods is photoactivation localization microscopy (PALM), which uses photoactivatable fluorescence proteins (PAFPs) with the characteristic of starting out in an “OFF” state that can be switched into a distinctly fluorescent “ON” state by irradiation with a specific wavelength (Figure 5D) [115]. Prior to fluorescence detection, a small subpopulation of FPs within the sample is converted from the “OFF” state to the “ON” state by inefficient photoactivation. Afterwards an image is recorded, the “ON” state FPs are photobleached to switch off their fluorescence and the process is repeated. Individual images, which were fitted to the respective PSFs, are then combined into a composite representing the fluorescence signal of every detected FP with a single molecule resolution below the diffraction limit. The use of PALM for in vivo imaging of *Drosophila* is still rare but has been exemplified in detail for visualizing E-cadherin in embryonic tissue with the use of mEosFP1 [116]. Similarly, the mEos derived pcStar has been recently employed in another variant of SMLM called single molecule-guided Bayesian localization microscopy (SIMBA) to detect E-cadherin nanostructures during early *Drosophila* embryogenesis [117]. More recent developments into PAFPs have yielded OsO_4_ fixation-resistant variants like mEos4b or mEosEM for the use in correlative light and electron microscopy (CLEM) approaches [118,119].

The other main category of super-resolution imaging is known as the reversible saturable optical fluorescence transition (RESOLFT) family and includes stimulated emission depletion (STED) as well as ground state depletion (GSD) [120]. The common idea behind these techniques is to use a fluorophore which can reversibly be switched by light between two states of distinguishable fluorescent properties. Combined with a spatial intensity distribution of a light beam containing a zero-point, the existence of the fluorophore in one specific fluorescent state can in theory be confined to an arbitrarily small volume far below the diffraction limit. In practice, this volume is limited by the intensity of light which can reasonably be used in the context of biological tissue, setting a soft resolution limit at around 20 nm—roughly one order of magnitude below the one suggested by Abbe’s law. RESOLFT microscopy relies on reversibly photoswitchable fluorescent proteins (RSFPs) and has already been successfully applied to *Drosophila* larval and adult tissue. The previously reported rsEGFP2 was used as a transgenic fusion with α-tubulin in a still rare in vivo proof of concept of time-lapse super-resolution imaging [121]. rsEGFP2 can be switched from a non-fluorescent “OFF” state to a green fluorescent “ON” state by irradiation with 405 nm, while 488 nm are used for fluorescence excitation and to simultaneously switch rsEGFP2 back to the “OFF” state. Alternatively developed RSFPs come with high stability for long-term time-lapse image acquisition (Kohinoor2.0), far red-shifted wavelength properties (rsFusionRed3) or dual-color modes for pulse-chase setups and other more elaborate investigations (mIrisFP) [122,123,124]. A fairly extensive list of currently available RSFPs for RESOLFT microscopy has been published recently [125].

So far, we discussed intrinsically fluorescent proteins, i.e., FPs that form a fluorophore within the protein structure itself. Extrinsically fluorescent proteins, on the other hand, are genetically encoded engineered enzymes with the capability to covalently bind fluorogenic substrates. In recent years, there have been a couple of these FPs developed as protein-tags like Halo, SNAP/CLIP or TMP. These tags were quickly adopted as a visualization alternative and applied in numerous approaches including studies in *Drosophila* tissue (Figure 5E) [126]. The Halo-tag is based on a bacterial haloalkane dehalogenase reacting with labeled chloroalkanes, SNAP- and CLIP-tags are both modified human O^6^-alkylguanine DNA alkyltransferases cleaving benzylguanine and benzylcytosine derivatives, respectively, and the TMP-tag is derived from a bacterial dehydrofolate reductase, binding trimethoprim derived substrates. Extrinsically fluorescent proteins have similar advantages over antibody stainings as conventional FPs, i.e., high signal linearity, easier sample preparation, lower background, more precise signal localization and less issues with tissue penetration. At the same time, the photophysical properties of their organic dyes mostly exceed that of intrinsical FPs. Among many other studies, this was demonstrated in the context of a long-standing labeling problem regarding tightly packed microvillar spaces within rhabdomeres of *Drosophila* PRCs which are only partially accessible for antibodies [127]. Using TRPL::SNAP alongside the fusion proteins Rh1::GFP, TRP::GFP and TRPL::GFP, it was shown that repeatedly published crescent shaped antibody staining patterns of rhabdomeral proteins were an artifact of poor tissue penetration. This technical artifact could be avoided with both intrinsically and extrinsically fluorescent tags, ultimately leading to a more detailed description of TRPL’s internalization dynamics. With the goal to visualize expression patterns and subcellular localization of neurotransmitter receptors in the *Drosophila* brain, a recent work opted to combine fluorescent protein trap technology with rapid chemical labeling of Brp::SNAP—a widely used marker for neuropils—instead of time-consuming immunochemical stainings, in order to allow automated image processing of large sets of fluorescence images [128]. While extrinsically fluorescent proteins do require the incubation with fluorogenic substrates which limits their usability in live imaging setups, they are much more modular than intrinsically fluorescent proteins. A protein of interest that has been tagged with a self-labeling enzyme can afterwards be labeled with various substrates specifically optimized for the imaging setup. These substrates can have distinct fluorescence spectra, act as enzyme blocking agents in pulse-chase experiments, serve as anchors for affinity-based protein purification or even be used in super-resolution microscopy like STED or CLEM due to stochastic blinking or contrasting properties [129,130].

## 5. Concluding Remarks

Fluorescent proteins in combination with optimized live imaging microscopy techniques have undeniably helped to further cell biological investigations thus far and will continue to do so for the foreseeable future. Despite being deployed in this context for the first time more than 25 years ago, GFP remains the most abundantly utilized FP in many imaging methods of biological tissue today. This is also true for investigations within the model system of the *Drosophila* eye as the above chapters of this review can confirm. GFP has been optimized many times, modified in various ways and combined with sensors to allow, for example, detection of Ca^2+^ ions, voltage changes and phosphoinositides or to generate FPs that can be switched between different fluorescent states by light for usage in super-resolution microscopy. Recombinant expression of reporters and sensors based on intrinsic and extrinsic fluorescence requires the labour-intensive generation of transgenic organisms. However, we recommend taking on this endeavour in view of the many advantages of FPs over immuno- and conventional chemical stainings. Investing into appropriate genetically encoded fluorescence tools can speed up investigations manifold and enable microscopic techniques not possible with conventional immunochemistry. In *Drosophila*, more versatile and modular FP variants will allow to couple forward genetic screens with subcellular molecule tracing, specific knock-downs and visualization on the nanoscopic scale.

## Figures and Tables

**Figure 1 ijms-22-08930-f001:**
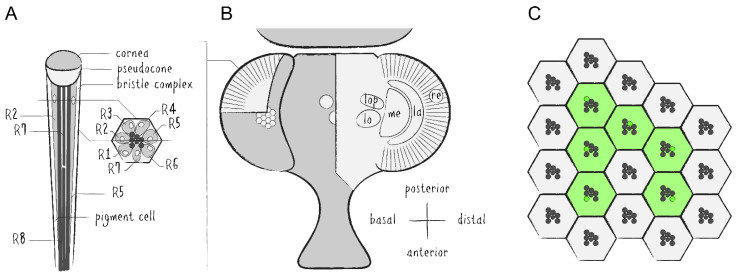
Schematics of the *Drosophila* visual system. (**A**) Illustration of a longitudinal section through the ommatidial center (left side) and a cross section through the plane just below the pseudocone, where the nuclei (white) of the photoreceptor cells (PRCs) R1–7 (light grey) are located (right side). Trapezoidal layout of the outer PRCs’ rhabdomeres in the center of the ommatidium is indicated (dark grey). Primary and secondary pigment cells not drawn. (**B**) Top view of a partially cross sectioned *Drosophila* head to illustrate the organization of the ommatidia (left side) and the optic lobe (right side). Indication of the hexagonal tiling by individual corneas on the curved surface of the compound eye and partial cross section to illustrate the wedged shape of ommatidial units. Order of neuropils from distal to basal: retina (re), lamina (la), medulla (me), lobula (lo) and lobula plate (lop). (**C**) Illustration of the repetitive organization and identical orientation of ommatidia and PRCs in a top view of a section of the compound eye. Origin of the pseudopupil phenomenon is indicated by highlighting the fluorescence of individual rhabdomeres from adjacent ommatidia (green) which together generate the fluorescent ommatidial trapezoid when rhabdomeral FPs are expressed in R1–6.

**Figure 2 ijms-22-08930-f002:**
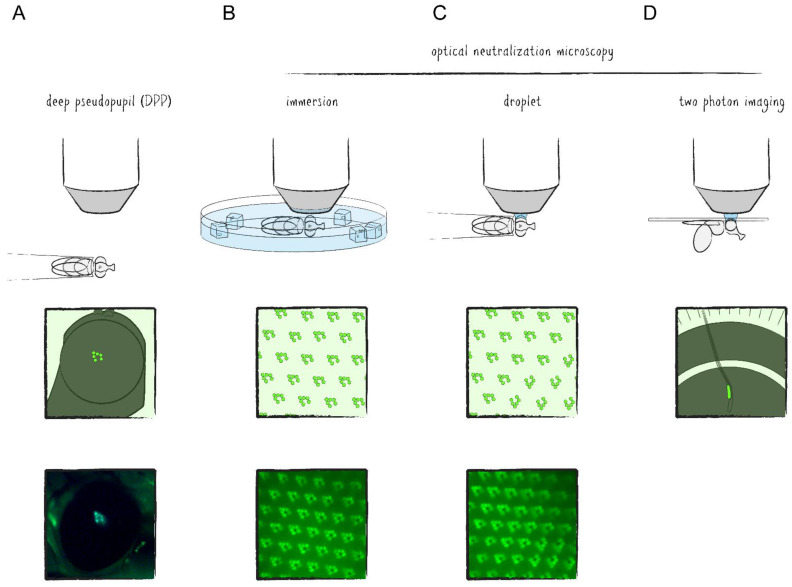
Illustration of common live imaging microscopy techniques of the *Drosophila* visual system. (**A**) A fluorescent deep pseudopupil can be visualized in anaesthetized or fixed flies by aiming a microscope objective (10×) slightly above or below the equator of the half spherical eye focusing approximately 180 µm below the cornea (top). (**B**,**C**) Optical cornea neutralization microscopy using water immersion and cold anaesthesia (indicated by ice cubes) or a single droplet between the eye of a fixed fly and the objective (20×) to detect fluorescent proteins in the rhabdomeres of ommatidia (top). (**D**) Imaging of the partially dissected optic lobe by two photon laser scanning microscopy (TPLSM) using a water droplet for optical neutralization (top). Illustration (center) and micrograph (bottom) of animals that express rhabdomerally located TRP::GFP in the outer photoreceptor cells (PRCs) R1–6 (**A**–**C**).

**Figure 3 ijms-22-08930-f003:**
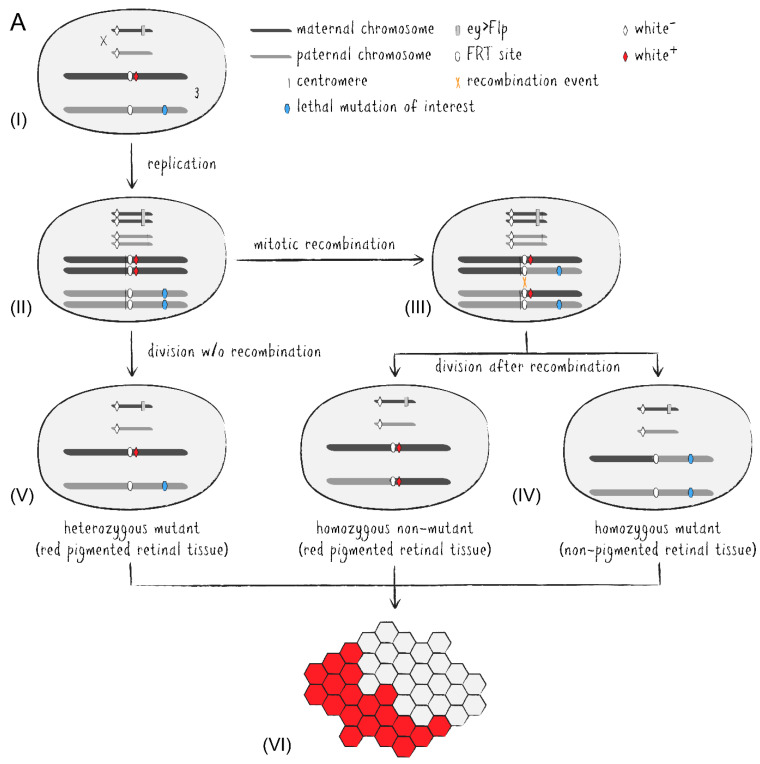
Schematic representation of mosaic tissue generation by mitotic recombination events via the Flp/FRT system adapted from *Saccharomyces cerevisiae*. (**A**) Conventional mosaic technique applied during *Drosophila* eye development utilizing pigmentation as phenotypic marker. (**I**) Heterozygous diploid cells with one maternal (dark grey bar) and one paternal (light grey bar) copy of chromatids (X and 3 illustrated, 2 and 4 not shown). Both X chromatids are mutant for the eye pigmentation gene *white* (white rhombus, *w^–^*). The yeast flippase gene is expressed eye-specifically from the maternal X chromatid by the enhancer of the *eyeless* gene (grey rectangle). In this example, both copies of the third chromatid carry a flippase recognition target (FRT) site (white oval) at the identical genomic locus of the same arm close to where the centromere will form. Distal on the same chromatid arm, the maternal copy carries a functional *white* gene (red rhombus, *w^+^*) as phenotypic cell clone marker, while the paternal copy carries the lethal mutation of interest (blue hexagon). (**II**) During development, chromatids are replicated into chromosomes tethered at the centromere (vertical line). (**III**) Flippase expression may lead to a recombination event (orange cross) between two homologous chromosome arms. (**IV**) If the recombination event exchanged a maternal for a paternal arm, cell division results in the generation of genetically distinguishable progeny. Cells receiving chromatids with two maternal arms are homozygously non-mutant for the gene of interest but carry both copies of *w^+^*, thus producing red pigmented retinal tissue (center). Cells receiving chromatids with two paternal arms are homozygously mutant for the gene of interest and carry no pigmentation marker, resulting in non-pigmented retinal tissue (right). (**V**) In the absence of a recombination event or in case of an exchange of identical sister chromatids, cell division generates heterozygously mutant progeny which also forms red pigmented retinal tissue due to the presence of one copy of *w^+^* (left). (**VI**) Adult eyes display mosaics of red and white ommatidia originating from *w^–^* and *w^+^* cell clones, respectively. White regions are homozygous for the lethal mutation of interest and can be investigated immediately adjacent to red pigmented non-mutant tissue. (**B**) Advanced variant for mosaic eyes in *Drosophila* utilizing two fluorescent proteins instead of eye pigmentation as phenotypic markers. The Gal4 transcription factor is expressed in outer photoreceptor cells by the Rh1 promoter (brown ribbon) and drives expression of both FP::NINAC fusion proteins under UAS control (green/red hourglasses). Homozygously mutant retinal tissue exclusively exhibits green fluorescence of rhabdomeres R1–6, while heterozygously mutant and homozygously non-mutant tissue displays green and red fluorescence simultaneously, appearing with yellow rhabdomeres in merge.

**Figure 4 ijms-22-08930-f004:**
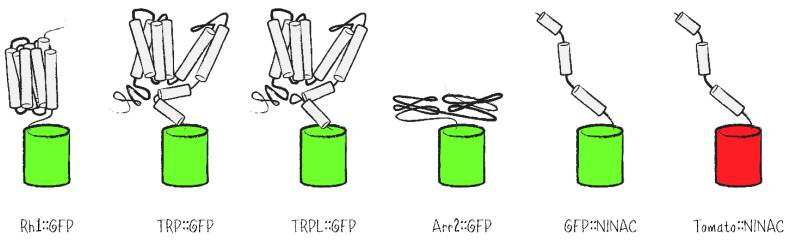
Schematic representation of common FP fusion proteins with applications in the research of the *Drosophila* visual system. Shown are FP fusions with rhabdomeral proteins (Rh1::GFP, TRP::GFP, TRPL::GFP, Arr2::GFP, GFP::NINAC, tomato::NINAC) for visualization of photoreceptor cells within the context of *Drosophila* mosaic eyes or as read-out for genetic screenings related to protein abundance, localization and trafficking as well as rhabdomere and cell integrity.

**Figure 5 ijms-22-08930-f005:**
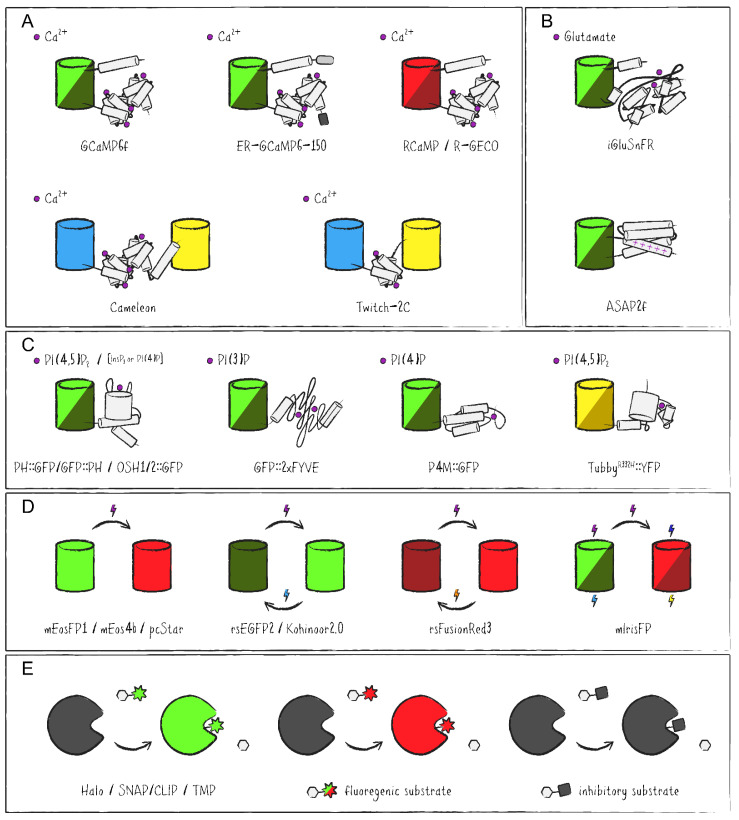
Schematic representation of fluorescent proteins (FPs) and FP fusion proteins with applications in the functional dissection of the *Drosophila* visual system. (**A**) Fluorescent genetically encoded calcium indicators based on conformational changes in singular circularly permuted (cp)FPs (GCaMP6f, ER-GCaMP6-150, RCaMP, R-GECO) or FRET probes (Cameleon, Twitch-2C) in response to Ca^2+^ binding. (**B**) Fluorescent biosensors for neuronal activity reacting to changes in extracellular glutamate concentration (iGluSnFR) or membrane potential (ASAP2f). Purple circles represent respective binding sites. (**C**) Phospholipid binding probes (PH::GFP, GFP::PH, OSH1/2::GFP, GFP::2xFYVE, P4M::GFP, Tubby^R322H^::YFP) for imaging intracellular pools of phosphoinositides like PI(3)P, PI(4)P, PI(4,5)P_2_ and InsP_3_. (**D**) Examples of advanced variants of FPs capable of photoconversion from green to red fluorescence (mEosFP1, mEos4b, pcStar), reversible photoswitching from an “OFF” state to an “ON” state (rsEGFP2, Kohinoor2.0, rsFusionRed3) or both (mIrisFP). (**E**) Extrinsically fluorescent protein tags able to covalently bind specific fluorogenic or inhibitory substrates allowing rapid chemical labeling (Halo, SNAP/CLIP, TMP).

## Data Availability

Not applicable.

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
