# Peer review of "Application of Fluorescent Proteins for Functional Dissection of the Drosophila Visual System"

_ijms, 2021, doi:10.3390/ijms22168930_

Round 1
Reviewer 1 Report
In this manuscript, the authors reviewed the applications of fluorescent proteins in studying the Drosophila visual system. The paper starts with a nice introduction talking about the importance of green fluorescent proteins and the Drosophila eye. Then it summarizes recent advances of using fluorescent proteins in forward genetic screens and as Ca2+ biosensors in the Drosophila visual system. Future directions of using fluorescent proteins in super-resolution imaging are also discussed. Overall, the manuscript is well-written and is a significant contribution to the field. I strongly recommend its publication in International Journal of Molecular Sciences.
A minor point for the authors to consider. In the Ca2+ biosensor part, recent development based on R-GECO has yield Ca2+ probes with longer emission and larger Stokes shift. Please cite and discuss the following articles:
Nat. Commun. 2014; 5: 5262.
Int. J. Mol. Sci. 2021, 22(1), 445
Author Response
We completely agree that the red fluorescent long Stokes shift variant REX-GECO1 represents an important development in the field of Ca2+ sensors especially with regard to two photon imaging and are happy to include it in our introductory section to our Ca2+ sensing chapter. Since REX-GECO1 has to our knowledge not been applied within the system of Drosophila, yet, we could not include it in any of the Drosophila-specific sections.
Reviewer 2 Report
The authors have summarized the application of fluorescent proteins for functional dissection of the Drosophila visual system. The article covers both structural imaging and function investigations. It is well organized with ample references. One neglect seems the lack of discussion regarding the potential implications of fluoresce on the visual system, given the nature of the subject studied. Nonetheless, the insight provided is helpful for FP researchers and greater community.
Author Response
The effect of fluorescent proteins on the visual system is certainly an interesting topic in itself. We are not aware of any particular study in Drosophila that investigated specifically this issue. From our own studies we know that fluorescence tags of TRPL generally don't appear to affect the activation of the phototransduction cascade which we typically show via electroretinography (Meyer et al., 2006, J Cell Sci; Schopf et al., 2019, J Histochem Cytochem). Beyond these data, however, we cannot say for sure that fluorescence has no influence on studies regarding visual perception. Due to lack of Drosophila studies we don't feel comfortable to include a speculative discussion about this topic in the current manuscript.